# Association between red cell distribution width and hypertension: Results from NHANES 1999–2018

**Ying Chen**[1], **Xiaoxiao Hou**[2], **Jiaxin Zhong**[3], **Kai Liu**[3]*

**1** Medical Laboratory Center, Hainan General Hospital, Hainan Affiliated Hospital of Hainan Medical University, Haikou, Hainan, 570311, China, **2** Department of Cardiology, Hainan General Hospital, Hainan Affiliated Hospital of Hainan Medical University, Haikou, Hainan, 570311, China, **3** Geriatric Center, Hainan General Hospital, Hainan Affiliated Hospital of Hainan Medical University, Haikou, Hainan, 570311, China

* hmliukai@hainmc.edu.cn

**Data Availability Statement:** Publicly available datasets were analyzed in this study. This data can be found here: https://www.cdc.gov/nchs/nhanes/index.htm.

## Abstract

The relationship between red cell distribution width (RDW) and hypertension remains a contentious topic, with a lack of large-scale studies focusing on the adults in the United States. This study aimed to investigate the association between RDW and hypertension among US adults from 1999 to 2018. Methods: Data were derived from the National Health and Nutrition Examination Survey (NHANES) 1999–2018. RDW values were obtained from the Laboratory Data's Complete Blood Count with 5-part Differential—Whole Blood module. Hypertension data were obtained through hypertension questionnaires and blood pressure measurements. Multivariable weighted logistic regression analyses were conducted to assess the association between RDW and hypertension, followed by subgroup and smooth curve analyses. Results: Compared to the non-hypertensive group, the hypertensive group exhibited higher RDW values (13.33±1.38 vs. 12.95±1.27, $P < 0.001$). After adjusting for covariates, weighted multivariable logistic regression analysis revealed a positive correlation between RDW and hypertension prevalence (OR: 1.17, 95% CI 1.13, 1.21, $P < 0.001$). When RDW was included as a categorical variable, participants in the fourth quartile had the highest risk of hypertension (OR: 1.86, 95% CI 1.70, 2.03, $P < 0.001$). Subgroup analysis showed that, except for age, BMI and weak/failing kidneys, gender, race, education level, smoking, alcohol use, congestive heart failure, and stroke did not significantly influence this correlation (all $P$-values for interaction >0.05).Smooth curve fitting analysis revealed a reverse J-shaped relationship between RDW and hypertension prevalence, with an inflection point at 12.93%. Conclusion: We first explored the relationship between RDW and hypertension among US adults and discovered a reverse J-shaped association, providing further insights into the relationship between blood cell counts and hypertension and offering a new foundation for hypertension prevention and control.

**Funding:** Project supported by Hainan Province Clinical Medical Center [2021276] The funder(Ying Chen)-Conceptualization, Data curation, Formal analysis, Writing – original draft

**Competing interests:** The authors declare no competing interests.

## Introduction

Hypertension, characterized by a systolic blood pressure (SBP) of $\geq$140 mmHg and/or a diastolic blood pressure (DBP) of $\geq$90 mmHg, poses a significant public health challenge in the United States [1]. With a prevalence ranging from 32% to 46% among adults, hypertension is a ubiquitous disease that profoundly impacts cardiovascular health [2–5]. Among preventable causes of cardiovascular disease-related mortality, hypertension ranks second only to smoking [6–9]. However, considering asymptomatic hypertension that remains undiagnosed, hypertension among adolescents, or adherence to the diagnostic criteria established by the American Heart Association in 2017, the estimated burden of hypertension in the United States is immense [10,11]. Therefore, the prevention and treatment strategies for hypertension in adults deserve primary attention.

Recent attention has shifted towards elucidating the potential association between red cell distribution width (RDW) and hypertension. RDW, a conveniently accessible biomarker through routine blood tests, has garnered significant interest as a potential indicator of cardiovascular health. Despite several studies exploring this link in diverse populations, the findings remain inconclusive and often contradictory. For instance, a case-control study conducted in northern Sudan, involving 156 subjects, failed to demonstrate a significant association between RDW and newly diagnosed hypertension in adults [12]. In contrast, another study encompassing 52 subjects reported a negative correlation between RDW and decreased systolic blood pressure (SBP) [13]. Additionally, a large-scale study conducted on a physical examination population in China revealed a reverse U-shaped relationship between RDW and hypertension, as well as systolic and diastolic blood pressure [14]. However, the research landscape pertaining to the relationship between RDW and the prevalence of hypertension among American adults remains relatively unexplored. There is a dearth of representative large-scale sample studies that comprehensively investigate this association. Given the significant impact of hypertension on cardiovascular health and the ease of obtaining RDW values through routine blood tests, it is imperative to fill this knowledge gap. Therefore, the rationale for performing this study is to provide insights into the potential association between RDW and hypertension among American adults, which could have significant implications for early diagnosis, prevention, and treatment strategies.

To address this knowledge gap, we conducted a comprehensive analysis utilizing data from the National Health and Nutrition Examination Survey (NHANES). Our objective is to elucidate the intricate relationship between red cell distribution width (RDW) and the risk of hypertension among adults in the United States. By gaining a deeper understanding of this association, our research aims to inform the development of early detection strategies and interventions targeting hypertension risk factors in the American adult population, ultimately contributing to the improvement of cardiovascular health outcomes.

## Method

### Ethics approval and consent to participate

The new ethic statement as follows: The NCHS Ethics Review Board protects the rights and welfare of NHANES participants. The NHANES protocol complies with the U.S. Department of Health and Human Services Policy for the Protection of Human Research Subjects. NCHS IRB/ERB Protocol Number or Description (https://www.cdc.gov/nchs/nhanes/irba98.htm). Ethical review and approval were waived for this study as it solely used publicly available data for research and publication. Informed consent was obtained from all subjects involved in the

NHANES. This study was deemed exempt from review by the Ethics Committee of Hainan General Hospital.

## Study population

Our study adopted a cross-sectional design and utilized data sourced from the National Health and Nutrition Examination Survey (NHANES), a comprehensive database tracking the health and nutritional status of the US population, updated biennially (https://www.cdc.gov/nchs/nhanes/index.htm). Employing a stratified cluster sampling method, we selected subject samples to ensure representativeness across diverse population groups. We included data spanning 10 cycles from 1999 to 2018, encompassing individuals with comprehensive information, including demographics, anthropometric measurements, blood pressure assessments and data, medical history, diabetes and kidney disease status, smoking and drinking habits, and results of complete blood counts(CBC). To uphold data quality and align with our research objectives, we excluded participants under 20 years old (n = 46,235), those pregnancy (n = 603), individuals missing CBC data (n = 10,274), non-respondents to the blood pressure questionnaire, and those missing blood pressure data (n = 51) from an initial pool of 101,316 participants. Ultimately, our analysis comprised 44,192 adults with complete information, as depicted in Fig 1, elucidating the registration process.

## Red cell distribution width measurement

RDW measures the range of variation in red blood cell volume, usually expressed as a percentage. A higher RDW value indicates greater variability in red blood cell size, while a lower RDW value suggests relatively uniform red blood cell sizes. Blood specimens collected during the National Health and Nutrition Examination Survey (NHANES) were processed and measured at the NHANES Mobile Examination Centers (MECs). The detailed procedures for specimen collection and processing are outlined in the NHANES Laboratory/Medical Technologists Procedures Manual (LPM). The methods used to derive complete blood count (CBC) parameters are based on the Beckman Coulter method of counting and sizing, in combination with an automatic diluting and mixing device for sample processing, and a single-beam photometer for hemoglobinometry. The white blood count (WBC) differential uses Volume -conductivity-light scattering (VCS) measurement technology. See Chapter 7 of the NHANES Laboratory/Medical Technologists Procedures Manual for details (NHANES 1999–2000 Procedure Manuals (cdc.gov)).

## Assessment of hypertension

Perform three or even four blood pressure measurements (systolic and diastolic blood pressure) during mobile examination centers (MEC) and home examinations of all eligible individuals using mercury sphygmomanometers. Participants aged 50 and above or under one year who are unable to travel to MEC will undergo a brief exam at home. Blood pressure measurement is performed by a MEC examiner. The technique used to obtain BP follows the latest recommendations of the American Heart Association Human Blood Pressure Determination by sphygmomanometers [15]. The blood pressure measurements of participants from 2017 to 2018 were conducted using the digital upper arm electronic measurement device Omron HEM-907XL, with three consecutive blood pressure measurements taken at 60 second intervals. Based on 2018 ESC/ESH Guidelines for the management of arterial hypertension [16] and recent literature on hypertension in NHANES, this study used four data sources to evaluate hypertension [17,18]. The first one is based on the questionnaire "Have you ever been informed by a doctor or other health professional that you have hypertension, also known as

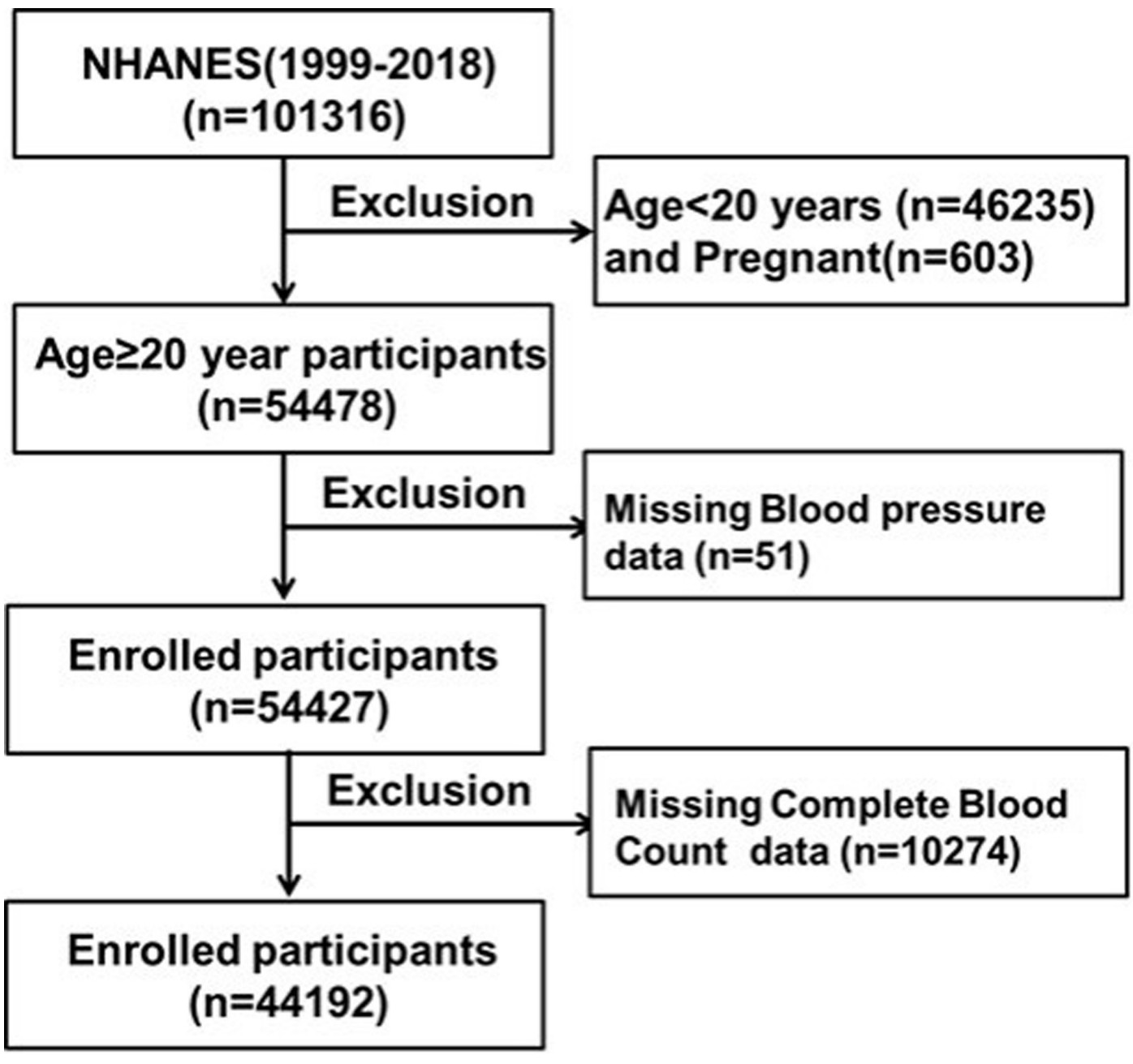

**Fig 1. Flowchart of participant enrollment process.**

hypertension?", with 15280 participants explicitly answering "Yes"; the second one is based on the questionnaire answering "No" or "Refused, Don't know, Missing", but with an average systolic blood pressure measured>140mmHg, with 3394 participants; The third one is the questionnaire answer No or Refused, Don't know, Missing, but the average diastolic blood pressure measured is>90mmHg, with 521 participants. Due to the questionnaire answer No or Refused, Don't know or Missing but with 0 participants receiving oral antihypertensive medication treatment, we used the first three types of hypertension data. A total of 19195 hypertensive participants were finally included in the study.

## Covariates

In our study, several potential factors that could influence the association between RDW and hypertension were carefully selected. These covariates include: Age (years), Gender (male/female),Race (Mexican American/ Other Race /Non-Hispanic White/Non-Hispanic Black), Body Mass Index (BMI, kg/m$^2$),Waist circumference, Education level(≤11th Grade, High School, College, College above), Family poverty income ratio (PIR),Smoking status(Yes/No), Alcohol use; Presence of chronic conditions: Chronic congestive heart failure, Coronary heart disease(Angina pectoris, Heart attack), Stroke, Weak/failing kidneys, Diabetes, Chronic bronchitis, Emphysema, Thyroid disease; laboratory parameters: White blood cell count (1000 cells/uL), Lymphocyte number (1000 cells/uL), Red blood cell count (million cells/uL), Hemoglobin (g/dL), Hematocrit (%), Red cell distribution width (%), Platelet count (1000 cells/uL), Mean platelet volume (fL), Blood Urea Nitrogen (mmol/L), Total Calcium (mmol/L), Serum Cholesterol (mmol/L), Creatinine (umol/L), Glucose (mmol/L), Triglycerides (mmol/L), Uric acid (umol/L), Sodium (mmol/L), Potassium (mmol/L), LDL-cholesterol (mg/dL), HDL-cholesterol (mg/dL) [18,19]. These covariates were chosen to comprehensively account for potential confounding factors and ensure a robust analysis of the association between RDW and hypertension in our study.

## Statistical analysis

Weights were generated in NHANES to accommodate the complex survey design, including oversampling, survey non-response, and post-stratification adjustments to align with total population counts from the Census Bureau. The calculation method for weights can be referenced in Supplementary Document 1. Continuous variables were expressed as mean ± standard deviation (SD) for normally distributed data or median (interquartile range) for skewed distributions. Normality was assessed using the Kolmogorov–Smirnov test. Categorical variables were presented as numbers (percentages). Baseline variable differences were evaluated using weighted t-tests, weighted chi-square tests, or Fisher's exact tests. For skewed continuous variables, between-group comparisons were performed using the weighted Wilcoxon rank-sum test.

A weighted logistic regression model was employed to explore the association between Red cell distribution width (RDW) and hypertension. Model 1 remained unadjusted for any covariates. Model 2 was adjusted for age and gender, while Model 3 was adjusted for Age, Gender, Race, BMI, Waist circumference, Education level, PIR, Smoking status, Alcohol use; Chronic congestive heart failure, Coronary heart disease (Angina pectoris, Heart attack), Stroke, Weak/failing kidneys, Diabetes, Chronic bronchitis, Emphysema, Thyroid disease; laboratory parameters: White blood cell count, Lymphocyte number, Red blood cell count, Hemoglobin, Hematocrit, Platelet count, Mean platelet volume, Blood Urea Nitrogen, Total Calcium, Serum Cholesterol, Creatinine, Glucose, Triglycerides, Uric acid, Sodium, Potassium, LDL-cholesterol, HDL-cholesterol.

Restricted cubic spline analysis (RCS) with three knots was applied to assess the nonlinear associations between RDW and hypertension, with the median value as a reference point. Two-piecewise logistic regression analysis models were utilized to examine the relationship between RDW and hypertension, identifying the inflection point. Subsequently, subgroup analysis categorized participants into different strata based on age, gender, race, education level, BMI, diabetes, coronary heart disease, stroke, smoking, alcohol use, and weak/failing kidneys, with added interaction terms to evaluate heterogeneity among subgroups. All statistical analyses were conducted using R software, version 4.3.1, with a significance level set at $P < 0.05$.

## Results

### Baseline clinical characteristics

Table 1 presents comparisons of demography and physical examination variables between the patients with hypertension (HTN) versus those without hypertension (Non-HTN). Patients with hypertension exhibited older age, slightly higher male proportion, elevated representation of Non-Hispanic White, lower educational attainment, higher rates of smoking and alcohol consumption, and increased prevalence of comorbidities including diabetes, congestive heart failure, coronary heart disease, stroke, chronic bronchitis, emphysema, and renal dysfunction. Additionally, hypertensive patients demonstrated significantly higher systolic and diastolic blood pressure, body mass index, and waist circumference compared to non-hypertensive individuals. However, there was no statistically significant difference in the Family power income ratio(PIR) between the two groups.

### Laboratory test results

The overall RDW value of the population is 13.11±1.34%. The hypertensive group is significantly higher than the non-hypertensive group (13.33±1.38 vs. 12.95±1.27%, $P<0.001$), Table 2. Notably, hypertensive individuals had higher white blood cell count, lower red blood cell count, hemoglobin, and hematocrit, higher blood urea nitrogen, total calcium, serum cholesterol, glucose, triglycerides, uric acid, and potassium levels. Additionally, they demonstrated increased mean platelet volume and lower platelet count. These findings suggest potential associations between hypertension and alterations in hematological and biochemical parameters.

### Association of RDW with hypertension

The weighted multivariate logistic regression analysis revealed a positive correlation between blood RDW and hypertension across all three regression models when RDW was treated as a continuous variable. After full adjustment for covariates, a significant association was observed, with an odds ratio (OR) of 1.17 and a 95% confidence interval (CI) of 1.13–1.21 (Table 3). Individuals in the third and fourth quartiles of RDW exhibited a higher risk of hypertension compared to those in the lowest quartile, as indicated by ORs of 1.45(1.35, 1.57) and 1.86(1.70, 2.03) in the fully adjusted model (Table 3). The relationship between RDW and hypertension demonstrated a reverse J-shaped curve according to the restricted cubic spline curve analysis (P for non-linear <0.001; Fig 2). The two-piece linear regression analysis identified a threshold effect for blood RDW on hypertension, with an inflection point at 12.93%. Below this threshold, each unit increase in RDW was associated with a lower risk of hypertension, as indicated by an odds ratio (OR) of 0.883 and a 95% confidence interval (CI) of 0.874–0.892 (P<0.001). Conversely, above the inflection point, each unit increase in RDW was associated with an increased risk of hypertension, with an OR of 1.117 and a 95%CI of 1.109–1.124 (*P*<0.001) (Table 4).

### Subgroup analysis

In order to further verify the stability of the correlation between RDW and hypertension in different populations, subgroup analysis was conducted. Interaction tests revealed no statistically significant differences in the correlation between RDW and hypertension among subgroups, except for age, weak/failing kidneys and BMI (<24.0/24.1–29.0/>29.1 kg/m$^2$) (Fig 3). This indicates that variables such as gender (male/female), race (Mexican American/Other Hispanic/Non-Hispanic White/Non-Hispanic Black/Other Race), education level (≤11th Grade/

**Table 1. Weighted baseline characteristics of participants.**

| Variables | Overall | Hypertension | Non-HTN | P |
|---|---|---|---|---|
| | n = 44192 | n = 19195 | n = 24997 | |
| Age(yr) | 49.79(18.18) | 60.08(15.54) | 41.89(15.97) | 0.025 |
| Gender(%) | | | | |
| Male | 21495(48.64) | 9453(49.25) | 12042(48.17) | 0.008 |
| Female | 22697(51.36) | 9742(50.75) | 12955(51.83) | |
| Race/Ethnicity (%) | | | | |
| Mexican American | 7965(18.02) | 2869(14.95) | 5096(20.39) | <0.001 |
| Other Races | 7161(16.20) | 2634(13.72) | 4527(18.11) | |
| Non-Hispanic White | 20132(45.56) | 9022(47.00) | 11110(44.45) | |
| Non-Hispanic Black | 8934(20.22) | 4670(24.33) | 4264(17.06) | |
| Education Level (%) | | | | |
| ≤11th Grade | 12340(27.92) | 6111(31.84) | 6229(24.92) | <0.001 |
| High School | 10173(23.02) | 4681(24.39) | 5492(21.97) | |
| College | 12299(27.83) | 5055(26.33) | 7244(28.98) | |
| College above | 9380(21.23) | 3348(17.44) | 6032(24.13) | |
| Smoking (%) | | | | |
| Yes | 20415(46.20) | 9625(50.14) | 10790(43.17) | <0.001 |
| No | 23777(53.80) | 9570(49.86) | 14207(56.83) | |
| Alcohol Use (%) | | | | |
| Yes | 28591(64.70) | 11974(62.38) | 16617(66.48) | <0.001 |
| No | 15601(35.30) | 7221(37.62) | 8380(33.52) | |
| Diabetes (%) | | | | |
| Yes | 5248(11.88) | 3973(20.70) | 1275(5.10) | <0.001 |
| No | 38944(88.12) | 15222(79.30) | 23722(94.90) | |
| Congestive heart failure(%) | | | | |
| Yes | 1451(3.28) | 1202(6.26) | 249(1.00) | <0.001 |
| No | 42741(96.72) | 17993(93.74) | 24748(99.00) | |
| Coronary heart disease(%) | | | | |
| Yes | 3365(7.61) | 2625(13.68) | 740(2.96) | <0.001 |
| No | 40827(92.39) | 16570(86.32) | 24257(97.04) | |
| Stroke(%) | | | | |
| Yes | 1654(3.74) | 1348(7.02) | 306(1.22) | <0.001 |
| No | 42538(96.26) | 17847(92.98) | 24691(98.78) | |
| Emphysema (%) | | | | |
| Yes | 912(2.06) | 632(3.29) | 280(1.12) | <0.001 |
| No | 43280(97.94) | 18563(96.71) | 24717(98.88) | |
| Thyroid diseases(%) | | | | |
| Yes | 615(1.39) | 358(1.87) | 257(1.03) | <0.001 |
| No | 43577(98.61) | 18837(98.13) | 24740(98.97) | |
| Antihypertensive drugs (%) | | | | |
| Yes | 37633(85.16) | 16365(85.26) | 21268(85.08) | 0.619 |
| No | 6559(14.84) | 2830(14.74) | 3729(14.92) | |
| Chronic bronchitis(%) | | | | |
| Yes | 2551(5.77) | 1482(7.72) | 1069(4.28) | <0.001 |
| No | 41641(94.23) | 17713(92.28) | 23928(95.72) | |
| Weak/failing kidneys (%) | | | | |
| Yes | 1228(2.78) | 937(4.88) | 291(1.16) | <0.001 |

*(Continued)*

**Table 1.** (Continued)

| Variables | Overall | Hypertension | Non-HTN | P |
|---|---|---|---|---|
| | n = 44192 | n = 19195 | n = 24997 | |
| No | 42964(97.22) | 18258(95.12) | 24706(98.84) | |
| Systolic pressure, mmHg | 123.17(31.87) | 137.09(32.46) | 112.49(26.91) | <0.001 |
| Diastolic pressure, mmHg | 70.22(19.12) | 73.26(20.29) | 67.89(17.82) | <0.001 |
| Family poverty income ratio,% | 2.53(1.62) | 2.48(1.59) | 2.58(1.65) | 0.992 |
| Body mass index, kg/m$^2$ | 28.86(6.72) | 30.36(7.09) | 27.71(6.18) | <0.001 |
| Waist Circumference (cm) | 98.54(15.93) | 103.26(15.76) | 94.92(15.09) | <0.001 |

Mean ± SD for continuous variables: *P* value was calculated by weighted t test. % for categorical variables: P value was calculated by weighted chi-square test. HTN, hypertension.

High School/College/College above), smoking (Yes/No), alcohol use, congestive heart failure (Yes/No), and stroke (Yes/No) did not significantly influence this correlation (all *P*-values for interaction >0.05).

## Discussion

In this comprehensive survey involving 44,070 adult participants in the United States, we observed a significant association between the level of red cell distribution width (RDW) and the risk of hypertension. Further validation of these findings was conducted through in-depth subgroup analysis and interaction testing, revealing a similar trend in this association. The

**Table 2. Laboratory test results of the research population.**

| Variables | Overall | Hypertension | Non-HTN | P |
|---|---|---|---|---|
| | n = 44192 | n = 19195 | n = 24997 | |
| White blood cell count (1000 cells/uL) | 7.25(2.45) | 7.31(2.65) | 7.21(2.29) | <0.001 |
| Lymphocyte number (1000 cells/uL) | 2.15(1.26) | 2.13(1.50) | 2.16(1.04) | 0.153 |
| Red blood cell count (million cells/uL) | 4.67(0.51) | 4.63(0.53) | 4.69(0.50) | <0.001 |
| Hemoglobin (g/dL) | 14.13(1.55) | 14.03(1.57) | 14.20(1.53) | <0.001 |
| Hematocrit (%) | 41.67(4.39) | 41.47(4.47) | 41.83(4.33) | <0.001 |
| Red cell distribution width (%) | 13.11(1.34) | 13.33(1.38) | 12.95(1.27) | <0.001 |
| Platelet count (1000 cells/uL) | 253.13(68.06) | 250.19(70.93) | 255.39(65.68) | <0.001 |
| Mean platelet volume (fL) | 8.20(0.93) | 8.23(0.95) | 8.18(0.91) | <0.001 |
| Blood Urea Nitrogen (mmol/L) | 4.87(2.18) | 5.48(2.64) | 4.40(1.60) | <0.001 |
| Total Calcium (mmol/L) | 2.36(0.09) | 2.36(0.10) | 2.35(0.09) | <0.001 |
| Serum Cholesterol(mmol/L) | 5.08(1.10) | 5.13(1.14) | 5.04(1.06) | <0.001 |
| Creatinine(umol/L) | 74.3[61.9,88.4] | 79.6[64.5,96.4] | 70.7[61.9,84.9] | <0.001 |
| Glucose(mmol/L) | 5.65(2.17) | 6.11(2.56) | 5.30(1.73) | <0.001 |
| Triglycerides(mmol/L) | 1.36[0.90,2.08] | 1.51[1.02,2.27] | 1.24[0.82,1.92] | <0.001 |
| Uric acid (umol/L) | 323.03(86.38) | 343.27(90.16) | 307.49(79.96) | <0.001 |
| Sodium (mmol/L) | 139.15(2.39) | 139.20(2.61) | 139.11(2.20) | 0.638 |
| Potassium (mmol/L) | 4.00(0.35) | 4.02(0.39) | 3.98(0.32) | <0.001 |
| LDL- cholesterol(mg/dL) | 115.78(35.77) | 115.77(36.01) | 115.79(35.59) | 0.961 |
| HDL- cholesterol(mg/dL) | 52.71(15.98) | 52.27(16.37) | 53.05(15.66) | <0.001 |

Mean ± SD for continuous variables: *P* value was calculated by weighted t test. % for categorical variables: P value was calculated by weighted chi-square test. Median [interquartile range] for continuous variables: P value was calculated by Wilcoxon rank-sum test. HTN,hypertension.

**Table 3. Association between blood RDW and hypertension.**

|  | Model 1 |  | Model 2 |  | Model 3 |  |
|---|---|---|---|---|---|---|
|  | OR(95%CI) | P | OR(95%CI) | P | OR(95%CI) | *P* |
| Continuous | 1.26 (1.23, 1.29) | <0.001 | 1.26 (1.23, 1.29) | <0.001 | 1.17 (1.13, 1.21) | <0.001 |
| Categories |  |  |  |  |  |  |
| Q1 | Ref. |  | Ref. |  | Ref. |  |
| Q2 | 1.42(1.32, 1.52) | <0.001 | 1.42(1.32, 1.52) | <0.001 | 1.23(1.14, 1.31) | <0.001 |
| Q3 | 1.89(1.78, 2.02) | <0.001 | 1.89(1.78, 2.02) | <0.001 | 1.45(1.35, 1.57) | <0.001 |
| Q4 | 2.62(2.45, 2.81) | <0.001 | 2.62(2.45, 2.81) | <0.001 | 1.86(1.70, 2.03) | <0.001 |
| P for trend |  | <0.001 |  | <0.001 |  | <0.001 |

Q1: 9.7–12.3%; Q2: 12.4–12.9%; Q3: 13.0–13.6%; Q4: ≥13.7%. RDW: Red cell distribution width.

relationship between RDW and the risk of hypertension displayed a reverse J-shaped; when RDW was below 12.93%, it was associated with a decreased risk of hypertension. However, once surpassing this threshold, RDW became associated with an increased risk of hypertension. These findings validate and extend the initial hypothesis proposed in this study, emphasizing the intricate interaction between RDW and cardiovascular health. This provides crucial insights into advancing our understanding of the pathophysiological processes in the field of cardiovascular-related diseases. For primary care physicians, the recognition of the link between RDW and hypertension risk offers a new tool in the early identification and management of hypertension. Routine blood tests, which typically include RDW measurements, can now be leveraged to assess an individual's hypertension risk. By monitoring changes in RDW levels over time, physicians can potentially identify patients who are at a higher risk of developing hypertension, enabling earlier interventions and prevention strategies. The inflection point identified in our study further enhances the clinical utility of RDW measurements. By knowing the threshold value where the risk of hypertension begins to increase, physicians can target specific patient subpopulations for closer monitoring and intervention. This personalized

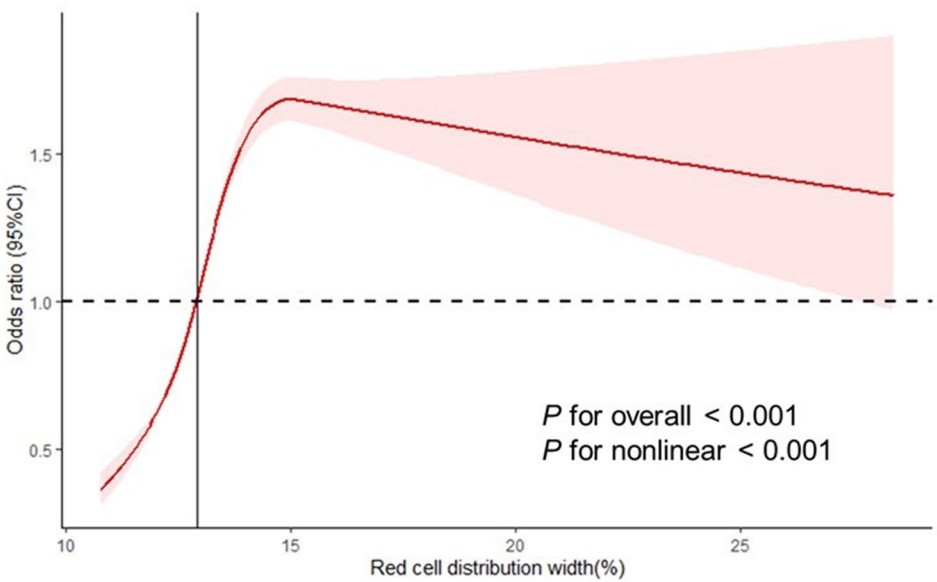

**Fig 2. Relationship between red cell distribution width and hypertension.**

**Table 4. Analysis of the threshold effect of blood RDW on hypertension by two-piece linear regression model.**

| Inflection point | OR(95%CI) | P |
|---|---|---|
| ≤12.93% | 0.883(0.874,0.892) | <0.001 |
| >12.93% | 1.117(1.109,1.124) | <0.001 |

approach to hypertension management can lead to improved patient outcomes and reduced healthcare costs.

The association between RDW and hypertension has been documented in the literature as early as 2014. Notable findings include a significant association between RDW and non-dipper hypertension [20,21], as well as pregnancy-induced hypertension [22]. In our study, we observed a substantial elevation in RDW levels in the hypertensive group compared to both non-hypertensive and overall participants, consistent with the results of the aforementioned studies. However, our research unveiled a non-linear relationship between RDW and hypertension, identifying a turning point at 12.93%. Below this threshold, a negative correlation was observed, whereas above it, a positive correlation was noted. This nuanced perspective provides a more accurate description of the relationship between RDW and hypertension. In a study primarily focusing on the Chinese population, after adjusting for age, white blood cell count, BMI, and HDL-C, a reversed U-shaped relationship was revealed, with RDW peaks for both females and males at 14.2% [14]. Interestingly, this inflection point closely resembles the conclusion of our study. Furthermore, another study emphasized that RDW values exceeding 14.0% were significantly associated with reduced red blood cell deformability, potentially impairing microcirculation blood flow [23].

Indeed, when RDW values drop below 12.93%, they demonstrate a negative correlation with adult hypertension prevalence. This association stems from several factors elucidated by recent research. Firstly, regarding Red Blood Cell Function, lower RDW values signify more uniformly sized red blood cells, indicating a healthier population with enhanced oxygen-carrying capacity and improved hemodynamics [24]. Consequently, optimal red blood cell function contributes to decreased blood pressure levels and a reduced risk of hypertension. Secondly, in terms of Inflammatory Status, reduced RDW values typically signal decreased systemic inflammation [25]. Chronic inflammation, a major contributor to hypertension, fosters endothelial dysfunction, arterial stiffness, and vascular remodeling [26]. Consequently, diminished inflammation levels correspondingly mitigate the likelihood of hypertension development. Lastly, with regard to Hematopoietic Health, RDW serves as a reflection of red blood cell size heterogeneity, influenced by factors such as nutritional status, bone marrow function, and erythropoietin levels [27]. RDW values below 12.93% suggest a healthier hematopoietic system characterized by more regulated red blood cell production. This optimal hematopoietic state bolsters overall cardiovascular health, thereby reducing hypertension risk.

Recent studies suggest that measuring RDW, an indicator of variations in red blood cell sizes, is emerging as a potential biomarker for various cardiovascular conditions, including hypertension [28]. Elevated RDW values above 12.93% may correlate with an increased risk of hypertension for several reasons. Firstly, high RDW levels often reflect increased inflammation and oxidative stress within the body, both pivotal in the development and progression of hypertension [29]. These processes impair vascular function, promote endothelial dysfunction, and contribute to arterial stiffness, all hallmarks of hypertension. Secondly, endothelial dysfunction, a common feature of hypertension characterized by impaired endothelial cell function along the blood vessels, is associated with elevated RDW levels [30]. This dysfunction can lead to compromised vasodilation and increased vascular resistance, ultimately contributing to elevated blood pressure. Thirdly, elevated RDW levels typically signify abnormalities in red

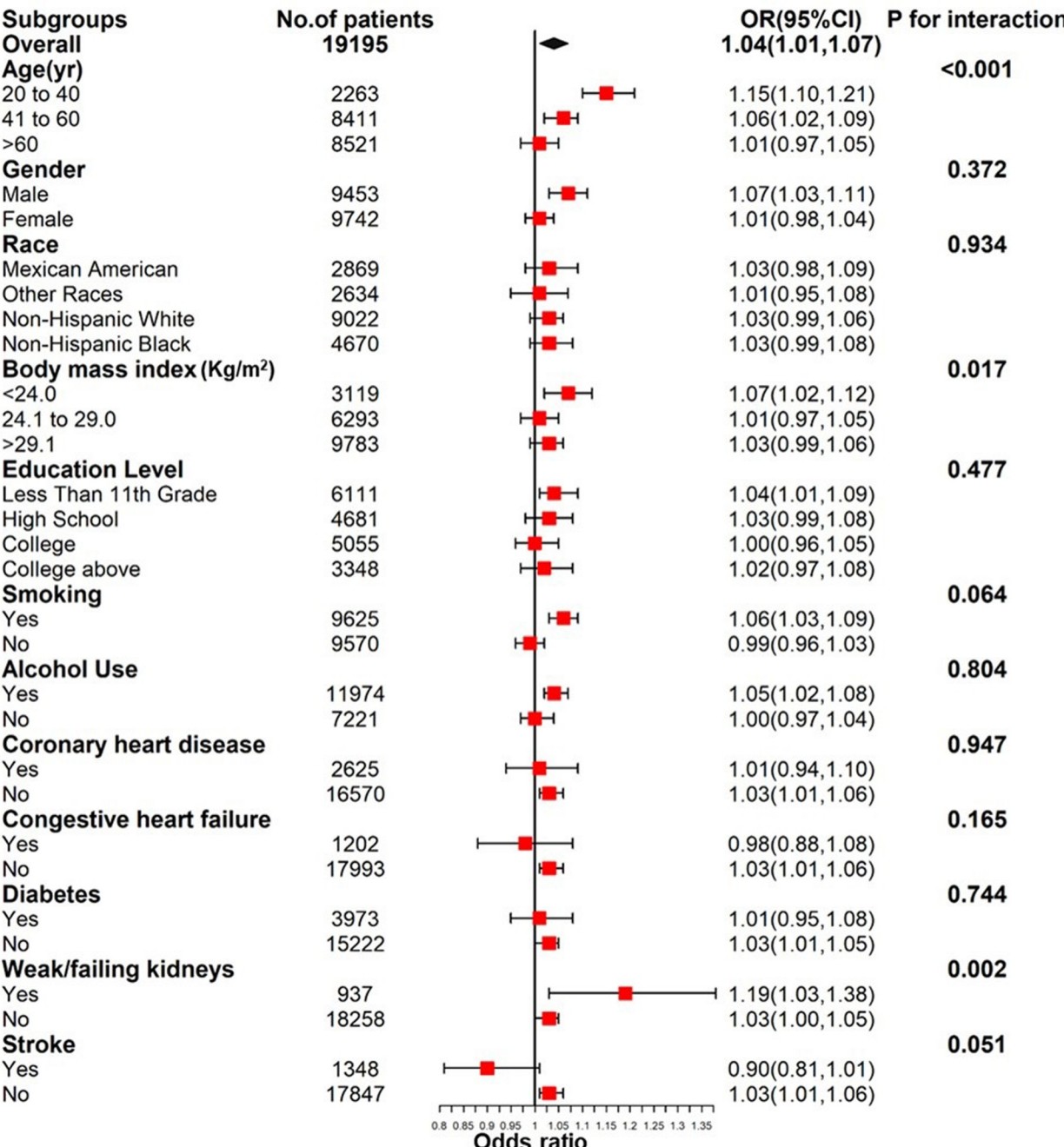

**Fig 3. Subgroup analysis for the association between red cell distribution width and hypertension.**

blood cell production, such as nutritional deficiencies or bone marrow disorders [31]. These anomalies indirectly contribute to hypertension by altering blood viscosity, impairing tissue oxygen delivery, and disrupting the balance of vasoactive substances. Fourthly, underlying chronic diseases such as chronic kidney disease [32], diabetes [33], and cardiovascular ailments [34], which often coexist with hypertension, can influence RDW levels and hypertension risk. RDW serves as a marker of overall health and disease severity in individuals with these conditions. Lastly, the positive correlation between elevated RDW levels and

hypertension may reflect shared underlying pathophysiological mechanisms, including chronic low-grade inflammation, endothelial dysfunction, and oxidative stress, which contribute to the development and progression of both conditions. In conclusion, elevated RDW levels (>12.93%) are associated with an increased risk of adult hypertension, likely due to their reflection of underlying inflammation, oxidative stress, endothelial dysfunction, hematologic abnormalities, and shared pathophysiological mechanisms. Monitoring RDW levels may offer insights for risk stratification and management of hypertension in clinical practice.

## Strengths and limitations

Compared to previous studies, this research holds significant advantages. Firstly, NHANES collects data from thousands of individuals every two years, and our study included 44,192 participants, making it the largest sample study to date examining the relationship between RDW and hypertension. This large sample size not only enhances the statistical power of the research but also improves the reliability and generalizability of the results. Secondly, NHANES employs a complex multi-stage sampling design, ensuring that participants come from diverse age, race, and ethnic groups, thus providing a representative sample of the non-institutionalized civilian population in the United States. This design makes the research findings more universal and applicable. Thirdly, NHANES has been collecting data regularly since the early 1960s, enabling us to analyze trends over a long period of time. This provides a valuable opportunity to assess the risk factors of hypertension and the effectiveness of intervention measures and policies. Fourthly, NHANES collects a wide range of health data, including physical measurements, laboratory tests, dietary intake, and physical activity. This comprehensive data collection allows researchers to comprehensively examine the relationship between health outcomes and various factors, effectively eliminating the interference of confounding factors on the conclusions. Finally, NHANES adopts standardized data collection protocols, ensuring consistency and accuracy during the survey, and reducing the risk of data bias. This helps us to more accurately compare data from different time periods and reveal health trends and changes.

While we have conducted a detailed analysis of the impact of RDW on hypertension prevalence in this study, there are several limitations that should be acknowledged. Firstly, this study solely utilized a nationally representative sample from the United States. Given significant racial differences in diet, physical activity, genetic variations, lipid metabolism, and cardiovascular disease susceptibility, the generalizability of our conclusions to other populations remains unclear. Secondly, due to the inherent nature of cross-sectional studies, establishing a causal relationship between RDW and hypertension poses challenges. Further prospective research is needed to ascertain the precise relationship between different forms of obesity and hypertension. Thirdly, despite adjusting for multiple covariates, we cannot entirely rule out the influence of other confounding factors on our results. Finally, the acquisition of information regarding hypertension medication treatment primarily relies on the "BPQ040A - Taking prescription for hypertension" section within the hypertension questionnaire. However, the lack of statistical analysis on all oral antihypertensive drugs may result in the omission of participants who were taking such medications due to recall bias, representing a limitation of this study.

## Conclusion

Our study findings demonstrate a close association between blood RDW levels and hypertension risk in US adults, revealing a reverse J-shaped relationship. This suggests that RDW may play a complex role in the pathogenesis of hypertension. We also explored the inflection point of RDW and differences in hypertension risk among different subgroups. Further exploration

of potential mechanisms is needed to better understand the role of RDW in the development of hypertension. We recommend larger-scale prospective studies to validate our findings, along with further experimental validation at the cellular and animal levels to elucidate the pathophysiological mechanisms underlying the association between RDW and hypertension.

## Supporting information

**S1 File.**
(PDF)

## Acknowledgments

We appreciate the NHANES database for providing a platform and contributors for uploading their meaningful datasets. We thank all participants in this study. Thank you to Min Zeng for his guidance on this article, and thank you to Tao Liu for his language revisions.

## Author Contributions

**Conceptualization:** Ying Chen.

**Data curation:** Ying Chen.

**Formal analysis:** Ying Chen, Xiaoxiao Hou.

**Funding acquisition:** Ying Chen.

**Resources:** Jiaxin Zhong.

**Software:** Xiaoxiao Hou, Kai Liu.

**Supervision:** Jiaxin Zhong, Kai Liu.

**Validation:** Xiaoxiao Hou, Jiaxin Zhong.

**Visualization:** Kai Liu.

**Writing – original draft:** Ying Chen, Xiaoxiao Hou, Jiaxin Zhong, Kai Liu.

**Writing – review & editing:** Kai Liu.

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
