## [Decision Letter · Decision Letter 0]

6 Feb 2024

PONE-D-23-40135Association between blood red cell distribution width and hypertension in U.S. adults: A cross-sectional study of 49622 NHANES participantsPLOS ONE

Dear Dr. Liu,

Thank you for submitting your manuscript to PLOS ONE. After careful consideration, we feel that it has merit but does not fully meet PLOS ONE’s publication criteria as it currently stands. Therefore, we invite you to submit a revised version of the manuscript that addresses the points raised during the review process.

We look forward to receiving your revised manuscript.

Kind regards,

Ming-Gang Deng

Academic Editor

PLOS ONE

Journal Requirements:

https://www.frontiersin.org/articles/10.3389/fcvm.2021.719165/full

In your revision ensure you cite all your sources (including your own works), and quote or rephrase any duplicated text outside the methods section. Further consideration is dependent on these concerns being addressed.

"Hainan Province Clinical Medical Center"

Reviewers' comments:

Reviewer's Responses to Questions

**Comments to the Author**

1. Is the manuscript technically sound, and do the data support the conclusions?

Reviewer #1: Yes

Reviewer #2: Yes

2. Has the statistical analysis been performed appropriately and rigorously? 

Reviewer #1: No

Reviewer #2: Yes

3. Have the authors made all data underlying the findings in their manuscript fully available?

Reviewer #1: Yes

Reviewer #2: Yes

4. Is the manuscript presented in an intelligible fashion and written in standard English?

Reviewer #1: No

Reviewer #2: Yes

5. Review Comments to the Author

Reviewer #1: 

The study is interesting to provide new research ideas for hypertension prevention by exploring the correlation between blood red blood cells and hypertension. However, there are still some questions that need to be answered further. The linguistic presentation of the manuscript needs to be reworked and consistency of methodology and results needs to be assured.

1. In Figure 1A, do the covariates not have any missing values, or are all missing values merged with missing red blood cell data. Please describe the other missing values of the covariates in the flowchart.

2. In Figure 1B, the diagnostic criteria for hypertension on the right-hand side and the "missing data" on the guide line need to be re-positioned so that they are not read ambiguously.

3. Does the diagnosis of hypertension include a history of medication use, if not please state in the deficiency

4. Covariates of smoking, alcohol consumption, chronic diseases are not mentioned in the methodology, please add and explain the specific categorization and relevant references.

5. Please confirm that all covariates are consistent with those in the methods description and in the presentation of results, e.g., family income, which appears only in the table of results but is not mentioned in the covariates or in the statistical methodology.

6. Red cell distribution width (RDW) needs to be defined or calculated with specific explanation in the method. "Q1: 9.7-12.4%; Q2: 12.5-12.9%; Q3: 13.0-13.7%; Q4: ≥13.8%" is the interquartile distribution of RDW or is it derived from other calculations, please specify in the methodology and cite the reference.

Reviewer #2: 

The authors elaborated the associations of RDW with hypertension. A reversed J-shaped relationship between blood RDW levels and the risk of hypertension was detected.  However, I still have some concerns:

1. The definition of hypertension used the thresholds as 140 mmHg for SBP and 90 mmHg for DBP. As your introduction suggested, the 2017 American Heart Association's diagnostic criteria was generally approved. Why not used the 2017 American Heart Association's diagnostic criteria in your study.

2. I noticed that some information lacked references, such as “Hypertension, defined as systolic blood pressure (SBP)≥140 mmHg and/or diastolic BP≥90 mmHg, is a widespread medical condition”. and some descriptions of covariates in the method section.

3. Many human biological indicators are not normally distributed. It is improper to used Mean (SD) and t test throughout the full-text.

4. It would be better to give the clinical practice potential applications based on your conclusions and some potential mechanisms between RDW and hypertension should be presented.

5. Please correct some spelling and vocabulary errors in the manuscript.

6. PLOS authors have the option to publish the peer review history of their article (what does this mean?). If published, this will include your full peer review and any attached files.

Reviewer #1: No

Reviewer #2: No

---

## [Author Response · Author response to Decision Letter 0]

5 Mar 2024

Dear editors and reviewers, 

Due to the fact that the article has almost been rewritten, it cannot be modified based on the original manuscript. We deeply apologize. The main reason for rewriting is that the original manuscript did not include weights for analysis, while NAHENS is a complex stratified sampling, so statistical analysis must include weights. There is a significant difference in the results between the two. And the discussion section is completely rewritten. I apologize on behalf of all authors for any inconvenience caused to the reviewers.

Here are the review comments and responses

Reviewer #1:

The study is interesting to provide new research ideas for hypertension prevention by exploring the correlation between blood red blood cells and hypertension. However, there are still some questions that need to be answered further. The linguistic presentation of the manuscript needs to be reworked and consistency of methodology and results needs to be assured.

1. In Figure 1A, do the covariates not have any missing values, or are all missing values merged with missing red blood cell data. Please describe the other missing values of the covariates in the flowchart.

Reply: Thank you for the guidance. Our data processing workflow begins with data merging, excluding variables with missing data exceeding 20% from the analysis. Subsequently, multiple imputation is applied to handle all missing data. The distribution of missing data is illustrated in the following diagram. Since participants with missing blood cell counts were excluded, red blood cell distribution width contains no missing values. For other covariates, including binary variables (Smoking, Alcohol Use, Diabetes, Congestive heart failure, Coronary heart disease, Stroke, Emphysema, Thyroid diseases, Antihypertensive drugs, Chronic bronchitis, Weak/failing kidneys) self-reported in the questionnaire as "Refused," "Don't know," or "Missing," a total of 39,878 instances were identified out of 574,509 records, accounting for 6.94% missing values. These missing values were imputed as "No." For BMI, Waist, PIR, White blood cell count, Lymphocyte number, Red blood cell count, Hemoglobin, Hematocrit, Red cell distribution width, Platelet count, Mean platelet volume, Blood Urea Nitrogen, Total Calcium, Serum Cholesterol, Creatinine, Glucose, Triglycerides, Uric acid, Sodium, Potassium, LDL-cholesterol, HDL-cholesterol, with a total data volume of 1,281,568 and 39,569 missing values (3.08%), multiple imputation was similarly applied to handle these missing values.

2. In Figure 1B, the diagnostic criteria for hypertension on the right-hand side and the "missing data" on the guide line need to be re-positioned so that they are not read ambiguously.

Reply: Thank you for the guidance. We have rewritten the definition of hypertension, referring to multiple recent NHANES based hypertension studies and providing a detailed description of the data acquisition process. The acquisition of hypertension data mainly comes from the Blood Pressure&Cholesterol (BPQ) questionnaire in NHANES and the Blood Pressure data in the Examination Data module.

3. Does the diagnosis of hypertension include a history of medication use, if not please state in the deficiency

Reply: Thank you for the guidance. We mainly register the information on medication treatment for hypertension through the "BPQ040A - Taking subscription for hypertension" section in the hypertension questionnaire. However, without statistical analysis of all oral antihypertensive drugs, some participants who took antihypertensive drugs may have missed recall bias, which is also a limitation of this study.

4. Covariates of smoking, alcohol consumption, chronic diseases are not mentioned in the methodology, please add and explain the specific categorization and relevant references.

Reply: Referring to previous research papers on NHANES, the acquisition of covariates such as Smoking, Alcohol Use, Diabetes, Congestive heart failure, Coronary heart disease, Stroke, Empire, Thyroid diseases, Chronic bronchitis, and Week/shipping kidneys is mainly achieved through the Smoking - Cigarette/Tobacco Use - Adult, Alcohol Use, and Medical Conditions data boxes in the Questionnaire Data module.

5. Please confirm that all covariates are consistent with those in the methods description and in the presentation of results, e.g., family income, which appears only in the table of results but is not mentioned in the covariates or in the statistical methodology.

Reply: Thank you for the reviewer's correction. The covariates have been double checked and the entire text is consistent.

6. Red cell distribution width (RDW) needs to be defined or calculated with specific explanation in the method. "Q1: 9.7-12.4%; Q2: 12.5-12.9%; Q3: 13.0-13.7%; Q4: ≥13.8%" is the interquartile distribution of RDW or is it derived from other calculations, please specify in the methodology and cite the reference.

Reply: Thank you for the reviewer's correction. The definition and detection method of red blood cell distribution width, in Method Red cell distribution width measurement RDW measures the range of variation in red blood cell volume, commonly expressed as a percentage A higher RDW value indicates greater variability in red blood cell size, while a lower RDW value suggests relatively uniform red blood cell sizes It has been described in detail in. Due to NHANES using independent data modules for each cycle, the detection methods for RDW in each cycle are listed.

The quartiles of RDW were calculated using our own original dataset, without reference to any literature.

Reviewer #2:

The authors elaborated the associations of RDW with hypertension. A reversed J-shaped relationship between blood RDW levels and the risk of hypertension was detected. However, I still have some concerns:

1. The definition of hypertension used the thresholds as 140 mmHg for SBP and 90 mmHg for DBP. As your introduction suggested, the 2017 American Heart Association's diagnostic criteria was generally approved. Why not used the 2017 American Heart Association's diagnostic criteria in your study.

Reply: Thank you for the corrections, reviewer. We utilized data from the period spanning 1999 to 2018, while the guideline version referred to is from 2017. Prior to 2017, the standard remained the same, with systolic blood pressure ≥140 mmHg being the criterion. Additionally, we referenced several published articles where this criterion is still applied (doi: 10.1186/s12199-021-01009-0; DOI: 10.1186/s12877-023-04191-8). Furthermore, the 2018 ESC/ESH Guidelines for the management of arterial hypertension also maintain the criterion of systolic blood pressure ≥140 mmHg (DOI: 10.1093/eurheartj/ehy339).

 2. I noticed that some information lacked references, such as “Hypertension, defined as systolic blood pressure (SBP)≥140 mmHg and/or diastolic BP≥90 mmHg, is a widespread medical condition”. and some descriptions of covariates in the method section.

Reply: Thank you for the corrections, reviewer. Diabetes, congestive heart failure, coronary heart disease, stroke, emphysema, thyroid diseases, antihypertensive drugs, chronic bronchitis, and weak/failing kidneys were all obtained from the NHANES database through questionnaire data. These covariates' definitions are clearly specified within NHANES.

3. Many human biological indicators are not normally distributed. It is improper to used Mean (SD) and t test throughout the full-text.

Reply: Creatinine and Triglycerides exhibit a significant skewed distribution, which has been modified in Table 1. Weighted comparison of inter group differences was performed using skewed statistical methods. Most other continuous variables exhibit a normal or approximately normal distribution.

4. It would be better to give the clinical practice potential applications based on your conclusions and some potential mechanisms between RDW and hypertension should be presented.

Reply: Thank you for the corrections, reviewer. The third and fourth paragraphs of the discussion section in the article analyze the potential mechanisms underlying the reverse J-shaped association between RDW and hypertension, with reference to a considerable body of literature.

5. Please correct some spelling and vocabulary errors in the manuscript.

Reply: Thank you for the reviewer's correction. The entire text has been revised. If the article is accepted, I am willing to pay a fee to an organization recognized by the magazine for language editing.

---

## [Decision Letter · Decision Letter 1]

2 Apr 2024

PONE-D-23-40135R1The association between red cell distribution width with hypertension: NHANES 1999–2018PLOS ONE

Dear Dr. Liu,

Thank you for submitting your manuscript to PLOS ONE. After careful consideration, we feel that it has merit but does not fully meet PLOS ONE’s publication criteria as it currently stands. Therefore, we invite you to submit a revised version of the manuscript that addresses the points raised during the review process.

We look forward to receiving your revised manuscript.

Kind regards,

Aleksandra Klisic

Academic Editor

PLOS ONE

Reviewers' comments:

Reviewer's Responses to Questions

**Comments to the Author**

1. If the authors have adequately addressed your comments raised in a previous round of review and you feel that this manuscript is now acceptable for publication, you may indicate that here to bypass the “Comments to the Author” section, enter your conflict of interest statement in the “Confidential to Editor” section, and submit your "Accept" recommendation.

Reviewer #2: All comments have been addressed

Reviewer #3: All comments have been addressed

Reviewer #4: (No Response)

2. Is the manuscript technically sound, and do the data support the conclusions?

Reviewer #2: Yes

Reviewer #3: Yes

Reviewer #4: Yes

3. Has the statistical analysis been performed appropriately and rigorously? 

Reviewer #2: Yes

Reviewer #3: Yes

Reviewer #4: Yes

4. Have the authors made all data underlying the findings in their manuscript fully available?

Reviewer #2: (No Response)

Reviewer #3: Yes

Reviewer #4: Yes

5. Is the manuscript presented in an intelligible fashion and written in standard English?

Reviewer #2: (No Response)

Reviewer #3: Yes

Reviewer #4: Yes

6. Review Comments to the Author

Reviewer #2: (No Response)

Reviewer #3: Chen et al. have performed a study on the association between RDW and hypertension in the US population. The study findings are interesting and they are well presented. I have some minor comments:

- The introduction section should be revised. The authors should focus on the main ideas related to the topic and try to emphasize the gaps in knowledge and the rationale for performing this study.

- A paragraph showing the clinical applications of these findings could be very helpful for readers.

Reviewer #4: The study titled "The association between red cell distribution width with hypertension: NHANES 1999–2018" is well-written with fair methodology. I have some comments:

- Define abbreviations in their first use and make sure you are using only abbreviated forms after the definition.

- Add the clinical utility and application of your findings for a primary care physician in the discussion section.

- Add the strengths of your study prior to the limitations paragraph.

- Upload figures with higher quality in the revised version.

- Add separate headings for "strengths and limitations" and "conclusions" sections.

- I found some typos and grammatical errors.

7. PLOS authors have the option to publish the peer review history of their article (what does this mean?). If published, this will include your full peer review and any attached files.

Reviewer #2: No

Reviewer #3: No

Reviewer #4: No

---

## [Author Response · Author response to Decision Letter 1]

8 Apr 2024

Reviewer #2: (No Response)

Reviewer #3: Chen et al. have performed a study on the association between RDW and hypertension in the US population. The study findings are interesting and they are well presented. I have some minor comments:

- The introduction section should be revised. The authors should focus on the main ideas related to the topic and try to emphasize the gaps in knowledge and the rationale for performing this study.

Response：Thank you to the reviewers and editors for their patient guidance.

The introduction section has been rewritten, with the first paragraph focusing on the current situation and hazards of adult hypertension in the United States. The second paragraph focuses on the current research status between the distribution width of red blood cells and hypertension, and explains the differences and shortcomings in the middle. The third paragraph elaborates on the necessity and importance of RDW and research on adult hypertension in the United States.

- A paragraph showing the clinical applications of these findings could be very helpful for readers.

Response：Thank you to the reviewers and editors for their patient guidance.

The first paragraph of the discussion has added a paragraph emphasizing the application value of this research result for primary care physicians.

Reviewer #4: The study titled "The association between red cell distribution width with hypertension: NHANES 1999–2018" is well-written with fair methodology. I have some comments:

- Define abbreviations in their first use and make sure you are using only abbreviated forms after the definition.

Response：The abbreviations for the entire text have been standardized, using the full name for the first occurrence and using abbreviations for subsequent occurrences.

- Add the clinical utility and application of your findings for a primary care physician in the discussion section.

Response：In the first paragraph of the discussion, the applicability of this study to primary healthcare physicians was added.

- Add the strengths of your study prior to the limitations paragraph.

Response：The research advantages have been increased, mainly in three aspects: research methods, research objects, and research results.

- Upload figures with higher quality in the revised version.

Response：High quality images have been uploaded.

- Add separate headings for "strengths and limitations" and "conclusions" sections.

Response：We have added separate headings for the sections "Advantages and Limitations" and "Conclusion" as required.

- I found some typos and grammatical errors.

Response：The entire text has been corrected for typing and grammar errors.

---

## [Decision Letter · Decision Letter 2]

23 Apr 2024

Association between red cell distribution width and hypertension: Results from NHANES 1999–2018

PONE-D-23-40135R2

Dear Dr. Liu,

We’re pleased to inform you that your manuscript has been judged scientifically suitable for publication and will be formally accepted for publication once it meets all outstanding technical requirements.

Kind regards,

Aleksandra Klisic

Academic Editor

PLOS ONE

Additional Editor Comments (optional):

Reviewers' comments:

Reviewer's Responses to Questions

**Comments to the Author**

1. If the authors have adequately addressed your comments raised in a previous round of review and you feel that this manuscript is now acceptable for publication, you may indicate that here to bypass the “Comments to the Author” section, enter your conflict of interest statement in the “Confidential to Editor” section, and submit your "Accept" recommendation.

Reviewer #3: All comments have been addressed

Reviewer #4: All comments have been addressed

2. Is the manuscript technically sound, and do the data support the conclusions?

Reviewer #3: Yes

Reviewer #4: (No Response)

3. Has the statistical analysis been performed appropriately and rigorously? 

Reviewer #3: Yes

Reviewer #4: (No Response)

4. Have the authors made all data underlying the findings in their manuscript fully available?

Reviewer #3: (No Response)

Reviewer #4: (No Response)

5. Is the manuscript presented in an intelligible fashion and written in standard English?

Reviewer #3: (No Response)

Reviewer #4: (No Response)

6. Review Comments to the Author

Reviewer #3: The authors have addressed all the comments by this reviewer and the manuscript is acceptable in its current form.

Reviewer #4: (No Response)

7. PLOS authors have the option to publish the peer review history of their article (what does this mean?). If published, this will include your full peer review and any attached files.

Reviewer #3: No

Reviewer #4: No

---

## [Editor Report · Acceptance letter]

30 Apr 2024

PONE-D-23-40135R2 

PLOS ONE

Dear Dr. Liu, 

I'm pleased to inform you that your manuscript has been deemed suitable for publication in PLOS ONE. Congratulations! Your manuscript is now being handed over to our production team.

Kind regards, 

on behalf of

Dr. Aleksandra Klisic 

Academic Editor

PLOS ONE